# Effects of Dietary Betaine on the Laying Performance, Antioxidant Capacity, and Uterus and Ovary Function of Laying Hens at the Late Stage of Production

**DOI:** 10.3390/ani13203283

**Published:** 2023-10-20

**Authors:** Feng Guo, Mengna Jing, Aaoyu Zhang, Jinfan Yi, Yanhong Zhang

**Affiliations:** College of Animal Science and Veterinary Medicine, Henan Institute of Science and Technology, Xinxiang 453003, China; guofeng@hist.edu.cn (F.G.);

**Keywords:** egg production, eggshell, reproductive organ, oxidation, methylation

## Abstract

**Simple Summary:**

Egg is a staple food in the human diet, containing all nutrients except vitamin C and dietary fibre. As the layers age, egg production and egg quality drop. Betaine is a natural compound widely found in many plants and animals, such as beets, spinach, wheat, and shellfish. It is readily available and cheap. Many studies have addressed betaine as a valuable dietary addition to alleviating inflammation, apoptosis, and oxidative stress. However, it is unclear whether dietary betaine consumption can improve the laying performance during the late-laying period. In this study, betaine was added to the diet of 65-week-old Jinghong-1 layers, whose laying performance declines after 60 weeks and can be below 80% at 80 weeks when the commercial cycle ends. We found that dietary betaine increased egg production and eggshell thickness. Our data further suggested that dietary betaine improved the function of the uterus and ovary, manifested by increased antioxidation capacity, higher expression of calcification-associated genes and hormone receptors, and decreased expression of pro-apoptotic genes. Considering the tendency to keep hens in production longer and the hens’ welfare, this study may offer a partial solution to the problem of egg production and quality.

**Abstract:**

Betaine has been found to alleviate oxidative stress, inflammation, and apoptosis. However, whether dietary betaine can protect late-laying hens against these adverse effects is unknown. Here, 270 65-week-old Jinghong-1 laying hens were randomly divided into the Control, 0.1% Betaine, and 0.5% Betaine groups and fed a basal diet, 0.1%, and 0.5% betaine supplemented diet, respectively. The trial lasted for seven weeks. Birds that consumed 0.5% betaine laid more eggs with thicker eggshells. Accordingly, uterine reduced glutathione (GSH), glutathione peroxidase (GSH-PX), and ovarian superoxide dismutase (SOD) contents were increased. The uterine calcium ion content and the mRNA expression of ovalbumin, ovotransferrin, and carbonic anhydrase two were increased. Moreover, ovarian IL-1β, Caspase-1, Caspase-8, and Caspase-9 mRNA expressions were decreased; luteinising hormone receptor (LHR) and follicle-stimulating hormone receptor mRNA expressions were increased. Furthermore, dietary betaine decreased the ovaries’ mRNA expression of DNA methyltransferase 1 (DNMT)1, DNMT3a, and DNMT3b. The methylation level at the promoter region of ovarian LHR decreased. These results indicated that dietary betaine consumption with a concentration of 0.5% could increase the laying rate and the eggshell thickness during the late-laying period. The underlying mechanism may include antioxidative, anti-apoptosis, and hormone-sensitivity-enhancing properties.

## 1. Introduction

Due to physiological changes, egg production and quality drop as hens age. Reproductive organs, including the ovary and oviduct, are the main targets of senescence. Ovary ageing is characterised by the apoptosis of the granulosa cells, chronic inflammation, oxidative stress, and abnormal hormonal responses [1,2]. These factors may join together to cause a decline in egg production. Concurrently, these factors were also found to contribute to the degradation of the uterus, where the eggshell is formed [3,4,5]. It is well known that the egg weight and eggshell surface area gradually increase when the hens become older. However, the eggshell becomes thinner, which causes an increase in cracked and broken eggs and further economic loss and food safety problems. Exploring efficient and practical ways to improve the egg-laying performance in the late-laying period is crucial to increasing the economic benefits and the welfare of laying hens.

Betaine, or trimethylglycine, a trimethyl derivative of glycine, is a natural compound which has been widely used in the poultry industry as a feed supplement. Many studies have shown that dietary betaine protected broilers against oxidative stress, metabolism disorders, and inflammation [6,7,8]. Attia et al. (2016) found that dietary betaine alleviated chronic heat-stress-induced decline in laying performance, manifested by improved laying rate, egg mass, and oviduct index [9]. A study by Omer et al. (2020) also showed that dietary betaine improved the egg-laying rate, accompanied by changes in hepatic lipid metabolism [10]. However, these studies mainly focused on the peak- instead of the late-laying period. The antioxidation and anti-inflammation properties of betaine suggest that it may be a potentially beneficial feed additive for aged hens. However, the effect of dietary betaine on the laying production and egg quality in the late-laying period and the underlying mechanism needs to be better revealed.

As an efficient methyl donor, betaine contributes methyl groups to the transmethylation cycle and can modulate DNA methylation, thereby regulating gene expression [11]. Zhao’s team found that dietary betaine regulated the expression of genes involved in the hens’ hepatic lipid metabolism through methylation modification [6,12]. Under a similar mechanism, dietary betaine also regulated the inflammatory gene expression in broilers [8,13]. DNA methylation is involved in ongoing animal ageing [14]. Recently, Cheng et al. (2023) found that the formation of low-quality speckled eggs was associated with methylation alteration in some ageing- and immune-related genes [15]. In the aged hens, ovarian DNA methylation changes in response to forced moulting may also contribute to the difference in egg quality [16]. However, whether dietary betaine could function through regulating DNA methylation to affect gene expression in reproductive organs in aged hens remains unknown. 

Therefore, this study aimed to investigate the effect of dietary betaine on the laying performance and egg quality of laying hens in the late-laying period. The uterine and ovarian antioxidative capacity and the mRNA expression of apoptotic factors, pro-inflammatory cytokines, and reproductive hormone receptors were further analysed to elucidate the underlying mechanisms.

## 2. Materials and Methods

### 2.1. Birds and Husbandry

Two hundred and seventy 65-week-old Jinghong-1 laying hens kept in the same middle layer of the battery cage were chosen to be involved in the study. Hens were weighed the day before the trial. Individuals with body weight out of the range of average ± 2SD were excluded, replaced by individuals with body weight within average ± 2SD. The newly joined hens were also from the middle layer of the battery cage. The first week was set as an adjustment period. The behaviour of the hens in the rearranged cages was closely watched to avoid aggressive behaviour. Hens involved were divided into three groups: the Control, 0.1% Betaine, and 0.5% Betaine groups, with six replicates of 15 birds each. The configuration of the cages was as shown in Figure 1. Feed troughs between different groups were separated to avoid the consumption of feed from other treatments.

The experiment lasted seven weeks, during which time birds in the Control group were fed a basal diet (Table 1), whilst their counterparts in the two betaine-treated groups were fed a basal diet supplemented with betaine (Betaine hydrochloride, 98% purity; Skystone Feed Co., Ltd., Yixing, China) at a dosage of 0.1% and 0.5%, respectively. The doses chosen were based on previous reports [10,17]. During the experimental period, water and diets were provided *ad libitum*. Daily egg number and weight were recorded for each replicate from the second week. The average value of eggs for each week was used to determine the laying rate (percentage hen/d) and egg weight. At the end of the trial (74 weeks), 30 eggs/group laid between 08:00 a.m. and 12:00 a.m. were randomly collected, and the egg shape index, eggshell thickness, white weight, yolk colour, and yolk weight were assessed. Evaluation of yolk colour was performed by using a Roche Color Yolk Fan. Six randomly chosen hens from each group were euthanised by decapitation. The uterus and ovary tissues were collected and frozen immediately in liquid nitrogen, then long-term stored at −80 °C. 

### 2.2. Measurement of Antioxidant Enzyme Activity

The concentration of malondialdehyde (MDA), reduced glutathione (GSH), glutathione peroxidase (GSH-PX), and superoxide dismutase (SOD) in the uterus and ovary tissue were detected by using commercial kits from Nanjing Jiancheng Bioengineering Institute (catalogue nos. A003, A006, A005, and A001) according to the manufacturer’s instructions.

### 2.3. Quantitation of mRNA Expression via Real-Time PCR

Uterus and ovary samples were homogenized by grounding in liquid nitrogen. Total RNA was then extracted using TRIzol reagent (15596026, Invitrogen, Carlsbad, CA, USA) from about 30 mg of sample. DNase I (D2215, Takara, Shiga, Japan) was used to eliminate genomic DNA contaminated with the total RNA. RNA quantification and integrity were analysed using a NanoDrop ND-1000 instrument and formaldehyde-containing agarose electrophoresis, respectively. cDNA was generated from 2 μg of total RNA using the GoScript Reverse Transcription System (A5001, Promega, Madison, WI, USA). Two microliters of diluted cDNA (1:15) were subjected to real-time PCR using TB Green Premix Ex Ta II (RR820A, TaRaKa, Shiga, Japan). The Primer Premier 5 software was used to design the primers, whose sequences are listed in Table 2. Chicken β-actin was used as a reference. Data derived from real-time PCR were processed using the 2^−ΔΔCt^ method and were presented as the fold change relative to the average level of the control group.

### 2.4. Methylated DNA Immunoprecipitation (MeDIP) Analysis

MeDIP analysis was conducted according to a previous report [13]. Briefly, about 40 mg of ovary tissue sample was ground in liquid nitrogen and digested with proteinase K dissolved in a lysis buffer (10 mM Tris-HCl, 25 mM EDTA, 0.1 M NaCl, and 0.5% SDS; pH 8.0) at 55 °C overnight. Genomic DNA was extracted using the phenol-chloroform isoamyl alcohol (PCI) method, precipitated with ethanol, and sonicated to obtain 200–700 bp fragments. One microgram of DNA fragments was denatured by bathing in boiling water for 10 min, followed by ice bathing. Five microliter antibodies (anti-5mC antibody; Abcam, ab10805, Boston, MA, USA and anti-5hmC antibody; Abcam, ab106918, Boston, MA, USA) were added to capture the methylated fragments. Meanwhile, 1 μg DNA fragments dissolved in 50 μL TE was reserved as the input control. After rolling the liquid sample on a mixer at 4 °C overnight, thirty microliters of Protein A/G PLUS-agarose (Santa Cruz, SC-2003, Dallas, TX, USA,) pretreated with denatured salmon sperm DNA were used to capture the antibody–nucleic acid complexes. Proteinase K was added to digest the antibodies and release the captured DNA. The DNA fragments were extracted using the PCI method and then subjected to real-time PCR. We obtained the information on the CpG islands in the promoter region by using two CpG island prediction software (CpGFinder and Urogene). The primers were designed with the Primer Premier 5 software; the information is shown in Table 3. Data were normalised to the input and presented as the fold change relative to the mean of the Control group.

### 2.5. Statistical Analysis

All statistical analyses were performed with SPSS 23.0 for Windows. All data were expressed as mean ± standard error. A one-way ANOVA followed by a post-hoc test (LSD) was applied to determine the effects of different dosages of betaine on laying rate and egg quality. An independent *t*-test was used to analyse the impact of 0.5% betaine on the mRNA expression and methylation modification. The level of significance in all analyses was set at *p* < 0.05.

## 3. Results

### 3.1. Betaine Increased the Egg-Laying Rate

The laying rate of birds in the Control and 0.1% Betaine group gradually decreased from week 2 to week 7. The laying rate in the 0.5% Betaine group slightly decreased from week 2 to week 5 but showed an upturn at week 6 and was significantly higher than that of the Control group at week 6 and week 7 (*p* < 0.05) (Figure 2A). The average egg weight in the 0.5% Betaine group was lower than that of the Control group during the experimental period (Figure 2B). 

### 3.2. Betaine Increased the Eggshell Thickness

The effect of betaine on the egg quality is shown in Table 4. Dietary betaine in a concentration of 0.5% significantly increased the eggshell thickness from 0.32 mm to about 0.42 mm (*p* < 0.01). However, the eggshell thickness in the 0.1% Betaine group was not changed compared with the Control group.

### 3.3. The Effect of Betaine on the Antioxidant Content in the Uterus and Ovary

The laying performance and egg quality results indicated that the dietary supplementation of betaine at a dose of 0.5% instead of 0.1% may play a positive role during the late-laying period. So, the subsequent studies were focused on the 0.5% Betaine group. The data in Table 5 show that consumption of 0.5% betaine significantly increased the content of GSH (*p* < 0.01) and GSH-PX (*p* = 0.05) in the uterus and SOD (*p* < 0.05) activity in the ovary. 

### 3.4. Dietary Betaine Consumption Increased the Ca^2+^ Concentration and Calcification-Associated Gene Expressions in the Uterus

As shown in Figure 3A, the concentration of calcium ions in the uterus of the hens consuming a 0.5% betaine diet was significantly higher than that in the Control group (*p* < 0.05). Betaine consumption also increased the mRNA expression of ovotransferrin, ovalbumin, and carbonic anhydrase 2 (CA2) in the uterus (*p* < 0.05) (Figure 3B).

### 3.5. Effect of Dietary Betaine on the mRNA Expression of Pro-Inflammatory Cytokines, Apoptosis Factors, and Reproductive Hormone Receptors

Dietary betaine consumption did not affect the mRNA expression of IL-1β, IL-6, and TNF-α in the uterus. However, it decreased the mRNA expression of IL-1β (*p* = 0.13) and conversely increased the mRNA expression of IL-6 (*p* < 0.05) in the ovary (Figure 4A).

Betaine consumption increased the mRNA expressions of ovarian luteinising hormone receptor (LHR) and follicle-stimulating hormone receptor (FSHR) (*p* < 0.05), but not that of oestrogen receptor (ER) β. Meanwhile, betaine consumption did not affect the mRNA expression of FSHR, LHR, and ERβ in the uterus (Figure 4B).

The mRNA expressions of the pro-apoptotic genes, Caspase-8 (*p* = 0.07), Caspase-9 (*p* = 0.08), and Caspase-3 (*p* = 0.16) in the ovaries of the hens that consumed 0.5% betaine were decreased compared with that in Control group. In the uterus, betaine consumption increased the mRNA expression of Bcl-2 (*p* = 0.08) (Figure 4C).

### 3.6. Effect of Dietary Betaine on the mRNA Expression of DNA Methylation-Associated Enzymes

As shown in Figure 5, betaine does not affect the mRNA expression of the major DNA methylation-associated enzymes in the uterus. Regarding the ovary, betaine significantly decreased the mRNA expression of DNA methyltransferases (DNMT)1, DNMT3a, and DNMT 3b (*p* < 0.05). Conversely, it increased the mRNA expression of DNA demethylase, ten-eleven translocation (Tet) 1 (*p* = 0.087) and Tet2 (*p* < 0.05).

### 3.7. Effects of Dietary Betaine on DNA Methylation of FSHR, LHR, Caspase-3, and Caspase-8 Gene Promoters in the Ovary

One CpG island was predicted at the promoter region of the LHR gene, but none were found at FSHR’s (Figure 6A,B). We then examined the 5mC and 5hmC levels on the CpG island at the promoter region of the ovarian LHR gene and along the 1000 bp of the 5′-flanking sequence of the ovarian FSHR gene. As shown in Figure 5E,F, dietary betaine significantly decreased the 5mC level at the promoter region of LHR (*p* < 0.01). However, betaine did not affect the 5mC level at the FSHR gene promoter and the 5hmC level at the two genes’ promoters.

One CpG island was predicted at the promoter region of the Caspase-3 and Caspase-8 genes, respectively (Figure 6C,D). The 5mC level at the promoter region of Caspase-3 was increased in the ovary of the birds consuming 0.5% betaine (*p* = 0.09). However, the 5mC level of the Caspase-8 promoter and the 5hmC level of the two genes’ promoters were unchanged (Figure 6G,H).

## 4. Discussion

Due to improvements in genetic selection and animal husbandry practices, industrial hens are kept longer. However, the eggs become bigger, the laying rate deteriorates, and the eggshell becomes thinner with hen age. In this study, the laying rate in the Control and 0.1% Betaine group gradually decreased throughout the trial. However, 0.5% betaine consumption lowered the egg weight and alleviated the decline in the laying rate. A previous study showed that dietary supplementation of betaine at a dose of 0.5% improves the laying performance of hen breeders at the peak-laying stage [10]; our results further support that dietary betaine with the same concentration could positively affect the laying rate during the late-laying period. 

It is impressive that dietary betaine increased the eggshell thickness remarkably. The eggshell thickness varies considerably among different breeds, laying periods, and diets. Jinghong-1 laying hens were reported to lay eggs with eggshells of 0.42 to 0.44 mm thick at the peak-laying period [18] and 0.35 mm at 64 weeks [19]. A previous study suggested that dietary betaine in a concentration of 0.15% increased the eggshell thickness of 2-year-old hens [17]. Betaine in a concentration of 0.12% increased the eggshell thickness of the eggs laid by 23-week-old Japanese quails [20]. Our result indicated that a higher dose of dietary betaine could increase the eggshell thickness during the late-laying period. Unfortunately, we did not examine the eggshell strength, which may more convincingly support the positive effect of betaine on eggshell quality. Nevertheless, previous studies addressed a positive relationship between eggshell thickness and strength [21]. It is worth noting that the significant increase in eggshell thickness may harm the aged hens, as it may cause calcium loss from the medullary and cortical bones, resulting in broken and weak bones [22]. Further research may be needed to reveal the effect of dietary betaine on bone status in older hens.

Chicken eggshell, a highly ordered and mineralised structure, mainly comprises calcium carbonate [23]. Many genes and proteins function in the mineralisation of the eggshell [24], and the dysfunction of those genes may cause a decrease in eggshell strength and thickness in the late-laying period [25]. Ovalbumin and ovotransferrin, two of the three major egg white proteins, play a role in the early stage of shell formation [26]. Ovalbumin was found capable of calcium-binding [27]. CA2, which exists in uterine glander cells, catalyses the reversible CO_2_ hydration and transports HCO_3_^−^ to the uterine fluid [28]. In vivo studies demonstrated that Ovocalyxin-32 (OCX32) is strongly associated with eggshell strength, thickness, and the termination of eggshell formation [29]. Previous studies showed that dietary microminerals, such as Se and cadmium, could affect the eggshell deposition by disrupting the expression of these genes related to eggshell mineralization [30,31]. To our knowledge, the mechanism underlying the effect of betaine on eggshell is still obscure. Our results indicated that betaine-promoted eggshell thickening might occur through influencing the early stage of eggshell formation instead of the terminal stage.

Due to their high egg yield, the reproductive organs of laying hens, especially older ones, are highly prone to oxidative stress, affecting the laying performance. Previous studies showed that betaine can protect broilers from oxidative stress [32,33]. However, the role of betaine on the antioxidative capacity of reproductive organs in laying hens is still unclear. Here in this study, dietary betaine with a dose of 0.5% increased the uterine GSH, GSH-PX, and ovarian SOD content, indicating a relatively lower level of oxidative stress and/or a higher antioxidative level. Researchers have tested many feed supplements regarding their benefits on egg-laying performance. Most of those studies found a coexistence of improved laying performance and lower oxidative stress levels [34,35]. Here, the results indicate that dietary betaine improves reproductive performance in the late-laying stage partially due to the alleviated oxidative stress in the reproductive organs.

In addition to oxidative stress, chronic inflammation, excessive apoptosis, and lower sensitivity to hormones are co-factors leading to the senesces of reproductive organs in the late-laying stage. Wang et al. (2020) and Grafl et al. (2017) found that oophoritis and salpingitis were the most prevalent pathomorphological changes in end-of-lay hens [36,37]. Feng et al. (2020) reported that aged (72 weeks) brown-laying hens showed higher IL-1β levels in the uterus compared with their younger (42-week) counterparts [38]. It has been previously found that many natural compounds supplemented to the diet of laying hens improved the reproductive organs’ function [4,35]. These mechanisms include alleviating the inflammation level by decreasing the expression of pro-inflammation cytokines such as IL-1β, IL-6, and TNF-α; reducing apoptosis by decreasing the expression of Caspase-3 and Caspase-8; and recovering the expression of reproductive hormone receptors, such as FSHR and LHR. In this study, dietary betaine consumption with a dose of 0.5% decreased the IL-1β expression but increased IL-6 expression in the ovary. Although IL-6 is well recognised as a pro-inflammatory cytokine, it is also suggested that a low and/or time-controlled release of IL-6 is associated with anti-inflammatory and antioxidant actions [39]. So, it is hard to conclude whether dietary betaine can relieve inflammation in the aged reproductive tract. Here, we also found that dietary betaine decreased the mRNA expression of Caspase-3, Caspase-8, and Caspase-9 and increased the mRNA expression of FSHR and LHR in the ovary, indicating that betaine reduced the apoptosis level and improved hormone sensitivity in the ovary, which may be one of the mechanisms by which betaine improves the laying rate. 

DNA methylation is catalysed by DNA methyltransferases (DNMTs), including DNMT1, DNMT3a, and DNMT3b, which transfer methyl groups from donors to cytosine residues on CpGs of gene promoters to regulate gene expression [40]. DNA can also be demethylated by ten-eleven translocation (Tet) protein, generating 5hmC, which is crucial in regulating gene expression [41]. Here, we found that betaine significantly reduced the expression of DNMT1, 3a, and DNMT3b and increased the mRNA expression of Tet2 in the ovary. However, none of these genes were affected in the uterus. We further demonstrated that the 5mC level in the promoter region of LHR was decreased. On the contrary, the 5mC level in the promoter region of Caspase-3 was increased. These changes in the 5mC level are consistent with the conclusion that hypermethylation in the promoter region consistently represses gene expression [40], which indicates that betaine affects the expression of LHR and Caspase-3 in the ovary through modulating the DNA methylation. Although betaine increased the mRNA expression of Tet1 and Tet2, we found no change in the 5hmC level in the promoter region of the genes detected, on which we can hardly draw any conclusion as the whole promoter region is examined. Further study using MeDIP-seq or the bisulfite sequencing method may help obtain a fuller picture of betaine’s effects.

## 5. Conclusions

We demonstrated that dietary betaine supplementation with a concentration of 0.5% increased the laying rate and the eggshell thickness of Jinghong-1 laying hens during the late-laying period. The underlying mechanism may include betaine’s antioxidative function, anti-apoptosis effect, and hormone-sensitivity-enhancing properties. Given the easy availability and low cost of betaine, as well as the tendency to keep hens longer in production, this study may provide a partial solution to the problem of egg production and quality.

## Figures and Tables

**Figure 1 animals-13-03283-f001:**
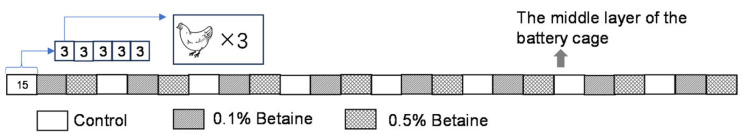
Schematic of cage configuration.

**Figure 2 animals-13-03283-f002:**
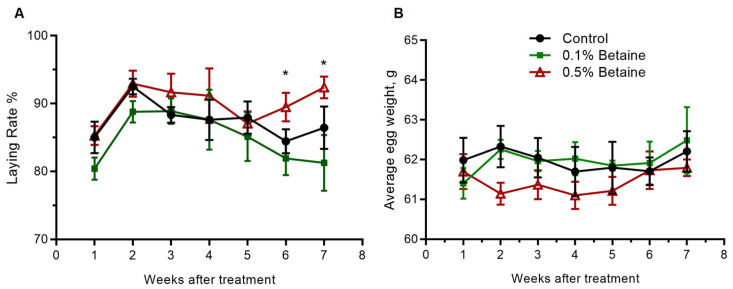
Mean egg-laying rate (**A**) and egg weight (**B**). Asterisk symbols denote significant differences (*p* < 0.05).

**Figure 3 animals-13-03283-f003:**
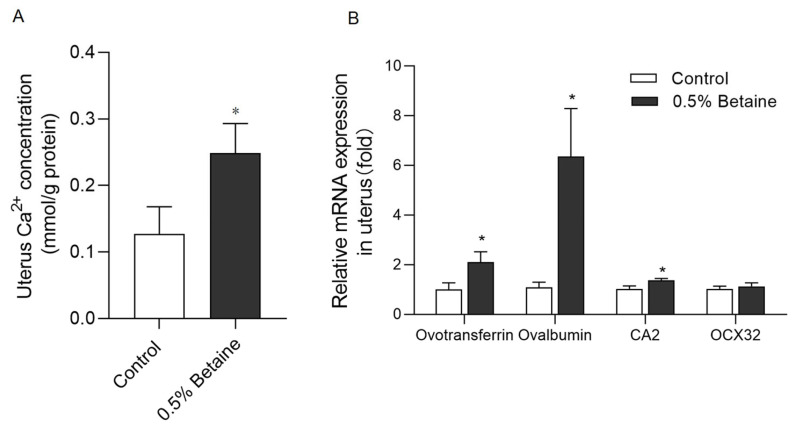
Effects of dietary betaine on the Ca^2+^ concentration (**A**) and calcification-associated gene expressions (**B**) in the uterus. Values are mean ± SEM, n = 6/group, Significance levels: * *p* < 0.05.

**Figure 4 animals-13-03283-f004:**
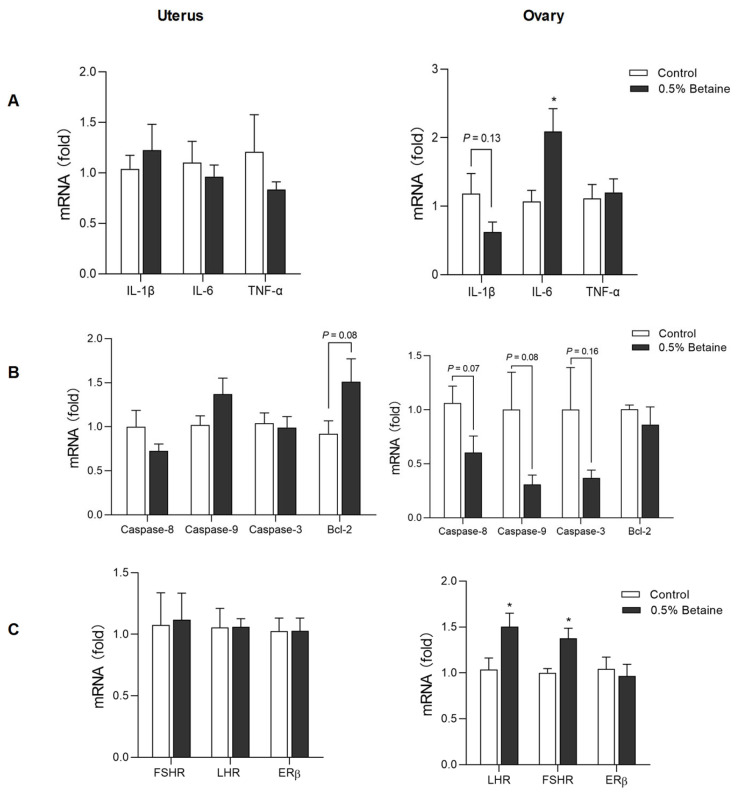
Effect of dietary betaine on the mRNA expression of pro-inflammatory cytokines (**A**), apoptosis factors (**B**), and reproductive hormones receptors (**C**) in the uterus and ovary tissues. Values are mean ± SEM, n = 6/group, Significance levels: * *p* < 0.05, *p* < 0.10 was used as the criterion for tendency.

**Figure 5 animals-13-03283-f005:**
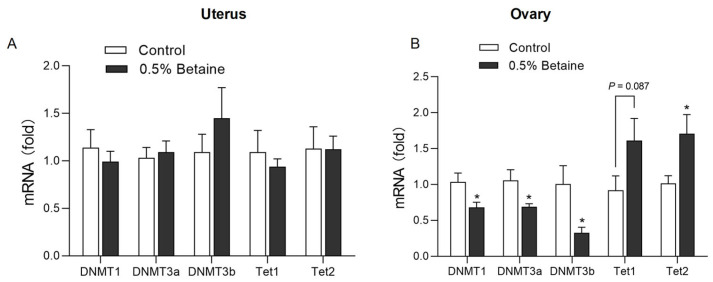
Effects of dietary betaine on the mRNA expression of DNA methylation-associated enzymes in the uterus (**A**) and ovary tissue (**B**). Values are mean ± SEM, n = 6/group, Significance levels: * *p* < 0.05, *p* < 0.10 was used as the criterion for tendency.

**Figure 6 animals-13-03283-f006:**
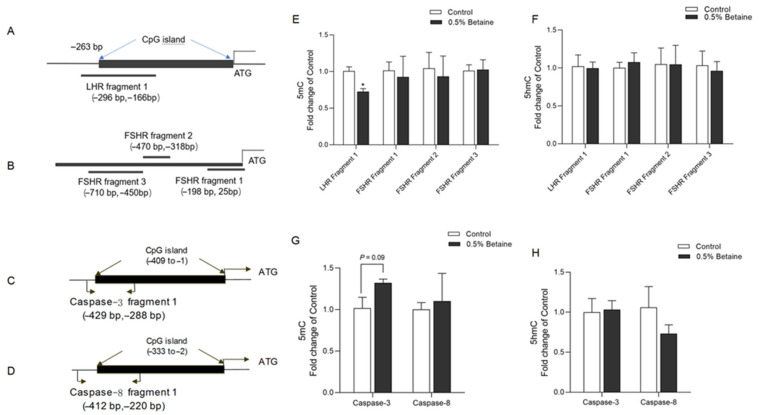
DNA methylation of the FSHR, LHR, Caspase-3, and Caspase-8 gene promoters in the ovary tissue. (**A**–**D**), Schematic diagram showing the amplified segments on the promoter sequence of LHR, FSHR, Caspase, and Caspase-8, respectively; (**E**,**F**), 5mC and 5hmC status on the promoter of LHR and FSHR, respectively; (**G**,**H**), 5mC and 5hmC status on the promoter of Caspase-3 and Caspase-8, respectively; Values are mean ± SEM, n = 4/group, Significance levels: * *p* < 0.05, *p* < 0.10 was used as the criterion for tendency.

**Table 1 animals-13-03283-t001:** Composition and nutrient levels of the basal diet.

Items	Content (%)
Cracked corn	61.90
Soybean meal	24.20
Wheat bran	1.00
Limestone	8.00
Dicalcium phosphate	1.50
Methionine	0.10
Sodium chloride	0.30
Premix ^1^	3.00
Total	100
Nutrient level ^2^	
ME (kcal/kg)	2584
CP	15.67
Lys	0.81
Digestible Lys	0.72
Thr	0.61
Digestible Thr	0.53
Met	0.26
Digestible Met	0.24
Calcium	3.31
Available phosphorus	0.32

^1^ The premix provided per kg of diet: Vitamin A, 10,000 IU; Vitamin D3, 2500 IU; Vitamin E, 50 mg; Vitamin K, 2.2 mg; Vitamin B1, 1 mg; Vitamin B2, 5 mg; Vitamin B6, 3 mg; Vitamin B12, 0.01 mg; niacin, 20 mg; folic acid, 1 mg; D-pantothenate, 6.6 mg; biotin, 0.12 mg; choline chloride (50%), 1000 mg; copper sulphate, 12 mg; ferrous sulphate, 90 mg; zinc sulphate, 70 mg; calcium iodine, 0.5 mg; manganese sulphate, 100 mg. ^2^ Values were calculated based on the tables of feed composition and nutritional values in China (31st Edition).

**Table 2 animals-13-03283-t002:** Primer sequence information (real-time PCR).

Target Gene	Sequence (F: Forward, R: Reverse, 5′-3′)	GeneBank Access
CA2	F: CTCCTCCGACAAGTCAGTGC R: TACGACGGCCAAACCATCAG	NM_205317.1
Ovotransferrin	F: TGACTTCCACCTCTTTGGGC R: GAATCCATCAGCGAGGGGAC	NM_205304.1
Ovalbumin	F: GAGTGGCATCAATGGCTTCTG R: RTCTAGGGCCATACCTGCTCAA	AH002466.2
OCX-32	F: AAGTCTCCGCCTGTAGTC R: CTTCCTTATCTGCTGCTTCA	NM-204534.5
β-actin	F: TGCGTGACATCAAGGAGAAG R: TGCCAGGGTACATTGTGGTA	NM-205518
TNF-α	F: TCACCCCTACCCTGTCCCA R: AGCCAAGTCAACGCTCCTG	NM-204267.2
IL-1β	F: TTCCGCTACACCCGCTCACA R: TGCCGCTCATCACACACGAC	Y15006
IL-6	F: GAAATCCCTCCTCGCCAATCTG R: GCCCTCACGGTCTTCTCCATAAA	AJ309540
LHR	F: GGTGTTCTCCATCCTGATAG R: TTGGCAATCTTGGTGTCTT	NM-205079
FSHR	F: ACCTTCCAAGCCTCAGATA R: TTAGCCGTAGAATCACACTT	EF621308
ERβ	F: ACCATTCAACGAAGCACTT R: ACTTCTAACCAGGCACATTC	Ab036415.1
Caspase-9	F: AGGGAGCAAGCACGACA R: GGTTGGACTGGGATGGAC	XM-424580.6
Caspase-8	F: AAGGAAGCGGGAAGAT R: GATACCTGAACGGAGACAC	NM-204184.1
Caspase-3	F: AAGGCTCCTGGTTTATTC R: CTGCCACTCTGCGATTTA	NM-204725.1
Bcl-2	F: CCGCATCCAGAGGGACT R: CGAAGAAGGCGACGAT	NM-205339.2
DNMT1	F: TTTTTTTACATAATCCTCCA R: AAAGTATCAATCCCCACTTG	NM_206952.1
DNMT3a	F: ATCACCACTCGCTCCAACTC R: CCAAACACCCTCTCCATCTC	NM_001024832.3
DNMT3b	F: TGATGAGAATCGCAGTAGA R: CACCAGGGAGAGTTAGAAA	NM_001024828.3
Tet1	F: AAAAGGAAGCGCTGTGAGAA F: CCACGCCAGTATGAGAATCA	XM_015278732.1
Tet2	F: CGGTCCTAATGTGGCAGCTA F: TGCCTTCTTTCCCAGTGTAGA	NM_001277794.1

**Table 3 animals-13-03283-t003:** Primer sequence information (MeDIP).

Target Gene/Fragment	Sequence (F: Forward, R: Reverse, 5′-3’)	Product (bp)
FSHR/Fragment 1	F: CTCCATCCTACTGTAAGCCCATAA	174
	R: GCAAGCAGGTGAGACCCAAG	
FSHR/Fragment 2	F: GATCTATGAAGGGGAGCAT	153
	R: TCTACAGTGGAAAGGGAGC	
FSHR/Fragment 3	F: GAGATCATTTGGGTTGTGG	261
	R: TTATGCTCCCCTTCATAGA	
LHR/Fragment 1	F: AGATCCACGCTTCCTCACAGTTTGGTG	131
	R: CGCTGTCCCTTCTGGAGGTGTTCACT	
Caspase-3/Fragment 1	F: TTTGGTGAGGCAATGTTACG	142
	R: TTTGGGCTCTGCGCTGTAAG	
Caspase-8/Fragment 1	F: CGGTAAGAGAAGAAAAGGAAAT	193
	R: GAGACTGAAGAAGTAGAAAATC	

**Table 4 animals-13-03283-t004:** Effects of dietary betaine supplementation on the egg quality of laying hens.

	Control	0.1% Betaine	0.5% Betaine
Eggshell thickness, 0.01 mm	32.0 ± 1.5 ^b^	32.3 ± 0.6 ^b^	41.6 ± 0.5 ^a^
Shape index, %	75.4 ± 0.49	76.4 ± 0.50	76.2 ± 0.47
Yolk weight, g	35.39 ± 0.59	36.8 ± 0.55	36.1 ± 0.49
Egg white weight, g	17.1 ± 0.27	17.4 ± 0.30	17.4 ± 0.28
Yolk colour	10.33 ± 0.17	10.7 ± 0.24	10.5 ± 0.18

Data are mean ± standard error of the mean (SEM) of 30 replicates per dietary treatment. ^a,b^ Means with different superscripts are different (*p* ≤ 0.05).

**Table 5 animals-13-03283-t005:** Antioxidant indices in the ovary and uterus.

Item	Control	0.5% Betaine	*p*
Uterus			
MDA (nmol/mg protein)	0.18 ± 0.07	0.25 ± 0.06	0.49
T-SOD (U/mg protein)	10.4 ± 0.71	8.7 ± 0.62	0.11
GSH (μmol/g protein)	65.17 ± 1.38	84.46 ± 3.03	0.001
GSH-PX (μmol/g protein)	61.96 ± 5.1	89.24 ± 11.01	0.05
Ovary			
MDA (nmol/mg protein)	0.95 ± 0.18	1.06 ± 0.11	0.63
T-SOD (U/mg protein)	14.69 ± 0.54	17.37 ± 1.06	0.04
GSH (μmol/g protein)	55.70 ± 7.88	53.10 ± 9.96	0.84
GSH-PX (μmol/g protein)	46.54 ± 1.13	45.90 ± 3.34	0.86

Data are mean ± SEM of six replicates per dietary treatment. MDA, malondialdehyde; T-SOD, total superoxide dismutase; GSH, glutathione; GSH-PX, glutathione peroxidase.

## Data Availability

All relevant data are within the manuscript.

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
