# Peer review of "Effects of Dietary Betaine on the Laying Performance, Antioxidant Capacity, and Uterus and Ovary Function of Laying Hens at the Late Stage of Production"

_animals, 2023, doi:10.3390/ani13203283_

Round 1

Reviewer 1 Report

Regarding the manuscript

Effects of dietary Betaine on laying performance, antioxidant capacity, and uterus and ovary function of laying hens at the late stage of production

This manuscript seems well designed

 I don’t see reasons to capitalize betaine in the title

L74-76 - Please improve the objective

The number of hens is not necessary

The major measurements should be indicated and the inclusion levels, since your conclusion was based on this information

L85 - Skystone Feed CO., Ltd,

Please review paper template to improve tables (especially font size and spaces)

L146 – analyze

Please explain how hens were selected before the trial starts

Do you think the small difference observed (even though not significant in your test) in egg weight influence the result?

Table 1 – please include nutrients under ingredients not in the side

Please include information of digestible Ly, Met, Thr

Inform the reasons to formulate diets with these requirements

 Since you have laying rate, egg weight different diets, please include feed intake and FCR results

  Discussion

I think you have to briefly indicate something about the betaine sources that are also available and contextualize

Also, about late stage, if we consider 65 + 7 weeks, there are many laying hens that can be reared longer than this. Please contextualize about the Jinghong-1 production cycle to the big audience

You just had 270 hens and 3 treatments, I would like to see the results for all treatments, not only control and 0.5% for most of the variables. Please include

Requires a minor revision

Reviewer 2 Report

The manuscript evaluates the effect of including Betaine in the diet of laying hens, birds that are in constant physiological oxidative stress. The idea is innovative and very well designed, clearly identifying the problem that exists, as well as technically very well justified due to the use of Betaine.

In general, the manuscript is very well written, with a clear methodology that allows for easy replication of this experiment by another group; basic principle of science. However, I will make some notes in order to improve the text.

1) in the abstract, avoid using acronyms without first writing out the name of the biomarkers.

2) Rewrite the objective; removing the methodology part of this section (example the number of birds). Remember that the objective will be used to complete; remembering that the conclusion must meet these objectives.

3) Avoid the word Betaine supplementation..... because that is not true; Supplementing is providing something more than the bird already consumes; which is not the case here. Review all text.

4) I did not find an explanation or reference for doses used in the methodology section. to review.

5) Table 1, include the chemical composition of the diet; preference for the measured; but if not, calculated is an option.

6) It was not clear in the text, because the authors tested two doses, but when presenting the results of some biomarkers they present data of only 0.5% Betaine.

7) Figure 2B, the caption is poor and the figure does not make it clear what is being measured; what gene?

8) Figure 3, the standard errors are extremely large; Does this data have a normal distribution? Have they been transformed? This information needs to be in the text.

9) I liked the writing of results and discussion, the conclusion could be stronger; less looking like results.
